# PIEZO1 and the mechanism of the long circulatory longevity of human red blood cells

**Simon Rogers**[1]*, **Virgilio L. Lew**[2]*

**1** School of Computing Science, University of Glasgow, Glasgow, United Kingdom, **2** Physiological Laboratory, Department of Physiology, Development and Neuroscience, University of Cambridge, Downing Site, Cambridge, United Kingdom

* simon.rogers@glasgow.ac.uk (SR); vll1@cam.ac.uk (VL)

**Data Availability Statement:** All relevant data are within the manuscript.

**Funding:** The background research work on which this investigation was based was supported over four decades by funding agencies in the UK and in

## Abstract

Human red blood cells (RBCs) have a circulatory lifespan of about four months. Under constant oxidative and mechanical stress, but devoid of organelles and deprived of biosynthetic capacity for protein renewal, RBCs undergo substantial homeostatic changes, progressive densification followed by late density reversal among others, changes assumed to have been harnessed by evolution to sustain the rheological competence of the RBCs for as long as possible. The unknown mechanisms by which this is achieved are the subject of this investigation. Each RBC traverses capillaries between 1000 and 2000 times per day, roughly one transit per minute. A dedicated Lifespan model of RBC homeostasis was developed as an extension of the RCM introduced in the previous paper to explore the cumulative patterns predicted for repetitive capillary transits over a standardized lifespan period of 120 days, using experimental data to constrain the range of acceptable model outcomes. Capillary transits were simulated by periods of elevated cell/medium volume ratios and by transient deformation-induced permeability changes attributed to PIEZO1 channel mediation as outlined in the previous paper. The first unexpected finding was that quantal density changes generated during single capillary transits cease accumulating after a few days and cannot account for the observed progressive densification of RBCs on their own, thus ruling out the quantal hypothesis. The second unexpected finding was that the documented patterns of RBC densification and late reversal could only be emulated by the implementation of a strict time-course of decay in the activities of the calcium and Na/K pumps, suggestive of a selective mechanism enabling the extended longevity of RBCs. The densification pattern over most of the circulatory lifespan was determined by calcium pump decay whereas late density reversal was shaped by the pattern of Na/K pump decay. A third finding was that both quantal changes and pump-decay regimes were necessary to account for the documented lifespan pattern, neither sufficient on their own. A fourth new finding revealed that RBCs exposed to levels of PIEZO1-medited calcium permeation above certain thresholds in the circulation could develop a pattern of early or late hyperdense collapse followed by delayed density reversal. When tested over much reduced lifespan periods the results reproduced the known circulatory fate of irreversible sickle cells, the cell subpopulation responsible for vaso-occlusion and for most of the clinical manifestations of sickle cell

the USA (VLL PI or co-PI). In the UK: Biotechnology and Biological Sciences Research Council (BB/E008542/1 9 (VL); BB/F001630/1 (VL), BB/F001673/1 (VL), and BB/H024867/1 (VL)), Engineering and Physical Sciences Research Council (EP/E059384 (VL)); The Wellcome Trust (064124; 061269; 059725; 030699; 033876; 15543; 17358; 13056 (all VL)), and The Medical Research Council (G8211073CA (VL)). In the US: NIH 2-RO1 HL28018-19 (VL); 2-RO1 HL21016-11 (VL).The funders had no role in study design, data collection and analysis, decision to publish, or preparation of the manuscript.

**Competing interests:** The authors have declared that no competing interests exist.

disease. Analysis of the results provided an insightful new understanding of the mechanisms driving the changes in RBC homeostasis during circulatory aging in health and disease.

## Author summary

The average circulatory lifespan of human red blood cells is about four months, amounting to about 200000 capillary transits. Among the many documented age-related changes red cells experience during this long sojourn the most relevant to homeostasis control comprise progressive densification with late density reversal, decline in the activities of calcium and sodium-potassium pumps, and slow inverse changes in their original sodium and potassium contents. Early experimental results have long established the view that these changes result from the cumulative effects of myriad capillary transits. However, many aspects of this process remain inaccessible to *in vivo* investigation. This prompted us to attempt a modelling approach applying a dedicated extension to our original red cell model. The results relegated the cumulative mechanism to a secondary role and exposed surprising critical roles for the declining patterns of the calcium and sodium-potassium pumps, as if harnessed by evolution to extend the circulatory longevity of cells within volume ranges that enable optimal rheological performance. The mechanism the model revealed implicated complex interactions between PIEZO1, the calcium-activated potassium channel KCNN4, the anion exchanger AE1, and the calcium and sodium-potassium pumps. These studies proved the model potential for exploring red cell homeostasis in health and disease.

## Introduction

Bridging the information gained from the study of single capillary transits [1] to the full RBC lifespan proved a formidable challenge, impossible to approach with repetitive dynamic state pages in the RCM. To undertake a proper lifespan study it became necessary to reset the core RCM into a new framework specifically dedicated to follow the homeostatic changes of RBCs throughout their long circulatory journey. This new framework was developed by one of us (SR) and labelled the Lifespan model. The nomenclature for PIEZO1-mediated permeabilities and other RBC variables and parameters are the same as those used and explained in the previous paper [1].

The Lifespan model enables detailed explorations of the multiple factors shaping the changes RBCs experience in the circulation. The simulations confirmed the versatility and potential of the Lifespan model as a dedicated tool for investigating the mechanisms that control the hydration condition of RBCs during circulatory aging in health and disease.

## Methods

### Open access to the Lifespan model

As for the core RCM, the Lifespan model, Lifespan*.jar, is available for downloading from the GitHub repository https://github.com/sdrogers/redcellmodeljava. The model operates as a *.jar programme within the JAVA environment which needs to be preinstalled. It is recommended not to alter the original file name as it contains coded information on date and update status. Operation of the Lifespan model generates *.csv files containing the results of

simulations with identical format to the one generated by the core RCM programme explained in detail in the comprehensive User Guide open to download from the GitHub repository together with the RCM*.jar and Lifespan*.jar programmes. The defaults set on the Lifespan model offered for download, if left unchanged, will automatically run the 120 lifespan trajectory of a "reference" RBC, the one found to represent best a realistic pattern of change. The "Ref" pattern responses are used for comparison with responses elicited by varying parameter values.

## Modelling the changes in RBC homeostasis during a standardized 120 day lifespan

The core RCM, framed into the new Lifespan model (Fig 1; details in the legend of Fig 1) allows testing a vast range of protocols using a dedicated user friendly interface. The operator sets the initial condition of the RBC, the parameter values under test, the duration of the simulation, the frequency of data outputs, and runs the model. While running, the user interface displays the changes in time of selected system variables in numerical and graphic formats for the full duration of the simulation. At the end of each simulation the full data sets of model variables can be saved in csv-file format. The variable names in the column headings are the same for both RCM and Lifespan models. The User Guide provides the units used and a full explanation of the function of the model variables and parameters.

The most relevant and widely supported experimental constraint guiding the model outcomes concerns the age-dependent progressive densification of RBCs throughout most of their circulatory lifespan. Recently matured RBCs with their full complement of haemoglobin are found at densities of about 1.080–1.085 g/ml, whereas aging RBCs are seldom found beyond densities of 1.105–1.110 g/ml [2–7] These boundaries then define a working range of age-related density progression in human RBCs from healthy donors for most of the RBC lifespan. A second important constraint for the model to comply with is the late density reversal phenomenon, originally reported by Borun et al., [2], overlooked for decades, and finally confirmed and extended in recent research [6,8,9]. Borun et al., [2] showed that between about 70 and 100 days in the circulation RBCs reverse their densification pattern, undergoing a partial, late and terminal density reversal. Progressive densification within established boundaries and late density reversal will be applied as the prime constraints on the parameter set for the model outcomes to comply with. Additional compliances with available evidence will be considered along the different stages of this investigation.

## Results and analysis

### Testing the quantal hypothesis

We investigated first whether the observed lifespan dehydration-densification patterns could be generated by the cumulative effects of single capillary transits, as suggested by the quantal hypothesis [10,11]. Fig 2 shows the predicted results at different PzCa levels. The levelling off trend shown for the first thirty days remained unchanged for 120 days. It can be seen that RBC densification increases with PzCa but only for the first 10–20 days, levelling off thereafter without any further volume reduction. Even at the highest PzCa of 80 h$^{-1}$, RBC density did not progress beyond the 1.090g/mL, a minimal dehydration level. Data analysis (csv files) of the volume (RCV) and net water fluxes (Fw) in the conditions with sufficient high PzCa to generated detectable dehydrations showed slow progress towards a fluid balanced condition with Fw approaching zero.

Interpreted in terms of the fluid balances analysed for single capillary transits (*in Fig 4 of the previous paper* [1]), the patterns in Fig 2 suggest that over the first twenty to thirty

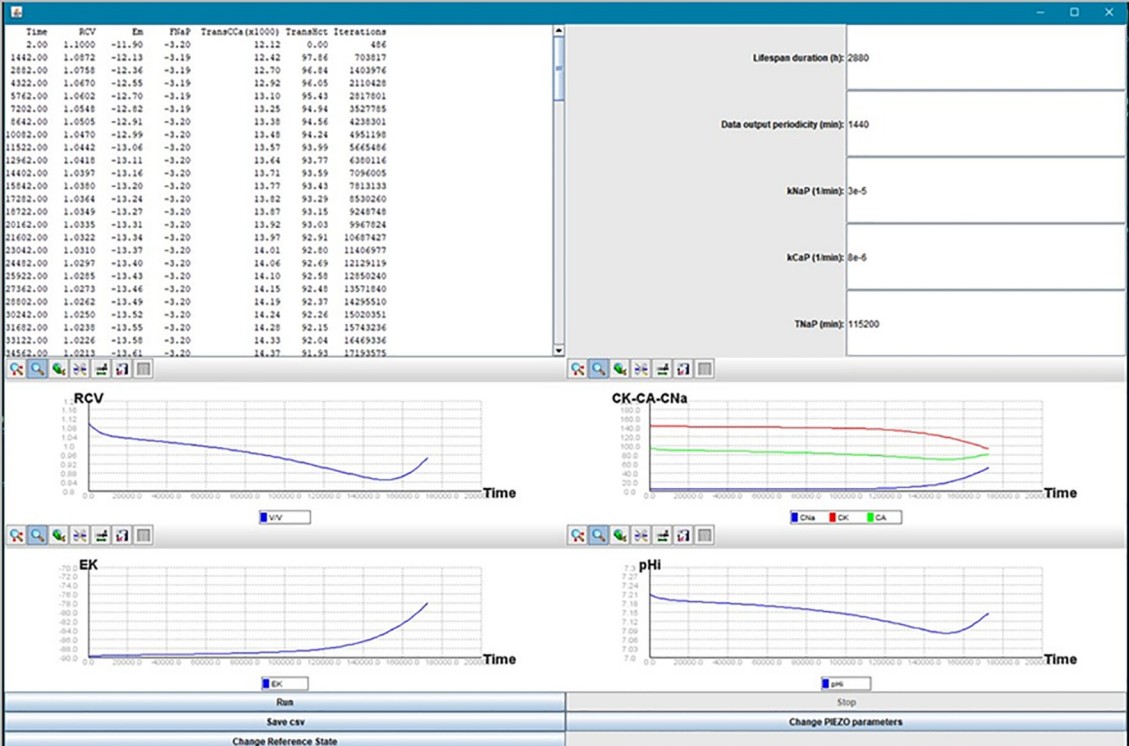

**Fig 1. User interface of the Lifespan model.** The interface was designed to offer information on the go during the computations allowing the model to be stopped, the protocol to be modified and the programme to be run again from the start. Running the Lifespan model with defaults generates the "Reference" output (Fig 3), analysed in detail in the text. The "Change Reference State" tag calls up a window with options for controlling the initial condition of the cell; the "Change PIEZO parameters" tag brings up the window for setting the PIEZO1-mediated permeabilities and the duration of the open-state. Three additional tags implement instructions for running (Run) and stopping (Stop) the computations and for generating csv files at the end of a run (Save csv) with the results displayed in the same format as that described before [1]. The main user interface is divided in three main panels: the top-right panel lists five time-dependent values, from top to bottom: Lifespan duration (h), data-output periodicity (min; defaulted to 1440, one output per day), rate constants of exponential decay for the Na/K (kNaP) and calcium pumps (kCaP; 1/min), and delayed onset time for Na/K pump decay (TNaP; min). In the running of the Lifespan model, selected data appear listed in the top left panel: Time (in min), Relative cell volume (RCV), membrane potential (Em, mV), Na/K pump-mediated Na efflux (FNaP, mmol/Loch), Trans-CCa (µmol/Loc), Trans-Hct (%), and number of model cycles in between data points (iterations). Trans-labelled variables report values at the end of the last capillary transit for the time point listed under Time. Selected variables are shown in graphic format in the bottom panel, clockwise from top left: RCV, CNa & CK & CA (mmol/Loc), pHi, and EK (the potassium equilibrium potential, mV). All default values for parameters and initial variables correspond to those used for the pattern defined by the Reference lifespan curve (Fig 3). The "Save csv" tag generates two csv files to enable comparisons between variables that change between the end of capillary transit (Transit-named csv file) and the end of intertransit periods for each time point: CVF/Ht, CCa, CCa2+, CH, MNa, MK, MA, MH, FCaP, FKGardos and all FzX ([1], Appendix]).

thousand transits, the initial volume surges caused by $CaCl_2$-influx during each of the brief PIEZO1 open states was followed by dehydration phases at progressively changing rates towards a state in which the fluid gained became exactly balanced by the dehydration level attained at the end of each intertransit period. Changing PIEZO1 parameters and model variables within admissible boundaries caused minor kinetic variations relative to those shown in Fig 2. But the two main features of the quantal response, non-progressive dehydration after an initial period, and minimal final dehydration levels even at high PzCa, were the consistent response.

These results rule out lifespan mechanisms based *exclusively* on cumulative effects of individual quantal changes during capillary passages. The quantal hypothesis tests were run with the implicit assumption that the kinetic properties of the main membrane transporters did not

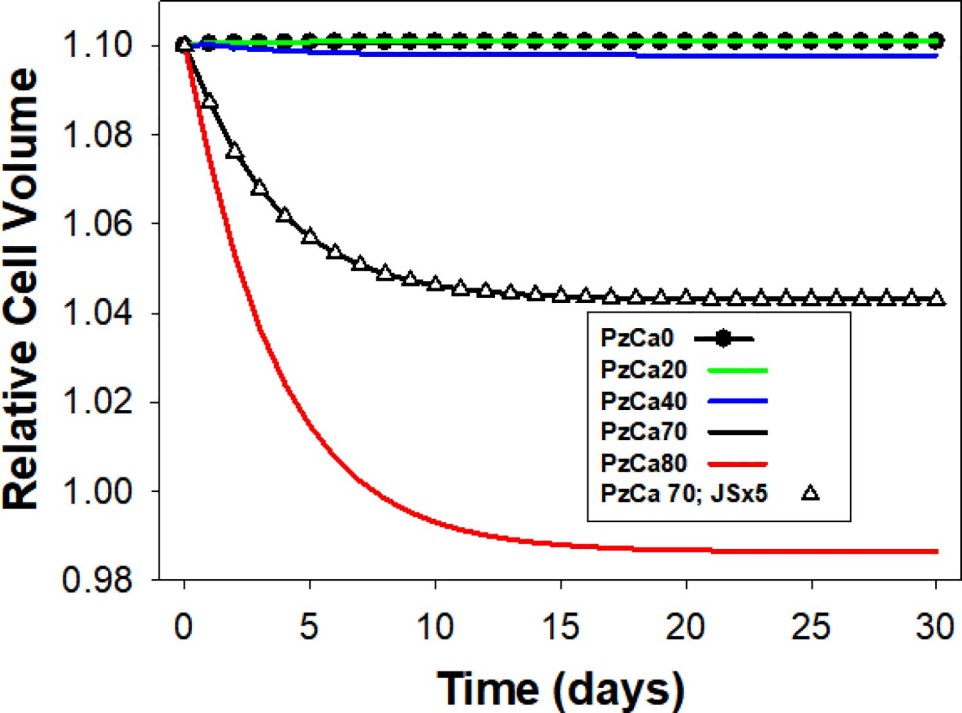

**Fig 2. Testing the quantal hypothesis; predicted changes in Relative Cell Volumes at different PzCa levels.** The trends shown only for the first thirty days remained unchanged for 120 days. Pump decay rate constants were set to zero. Triangles: effects of increasing JS turnover rate five-fold at PzCa = 70/h. Note that even at the highest PzCa shown, final dehydration does not exceed 10% of initial RCV.

change with cell age. This assumption is unrealistic. The decline in the activity of the calcium and Na/K pumps among other transporters is well documented [8,12–18].

## Analysis of pump-decay patterns

The measured distribution of $Ca^{2+}$-saturated PMCA calcium extrusion rates, Fmax, in RBCs from six different donors [12] rendered coefficients of variation (CV) of between 45 and 53%, with means to the right of the median in each, right skews ranging from 1.40 to 1.67. To represent such a decay process in the model we considered linear and exponential decay functions. The symmetry of linear decay distributions was incompatible with the observed skewed pattern. Exponential decay functions on the other hand generate right skews and were the obvious choice. There is no such detailed information on the modality of sodium pump decay. The few reliable findings document differences between light and dense cell fractions, substantial fall in $[H^3]$-ouabain binding and ouabain-sensitive potassium transport only in dense cell fractions [15,18], suggesting a late decline, unlike the gradual one of the PMCA. On this background, a large number of preliminary tests were carried out with the Lifespan model seeking to establish whether it was possible to generate a pattern in full compliance with the known densification-reversal sequence. Unexpectedly, the pattern that emerged from these trials and that fulfilled the expected compliance could be implemented with a very restricted set of pump-decay parameter values.

## The Reference pattern

We generated a protocol with default values which provides a standardized reference curve, the Reference pattern (Fig 3), that can be used for comparison with the effects of parameter

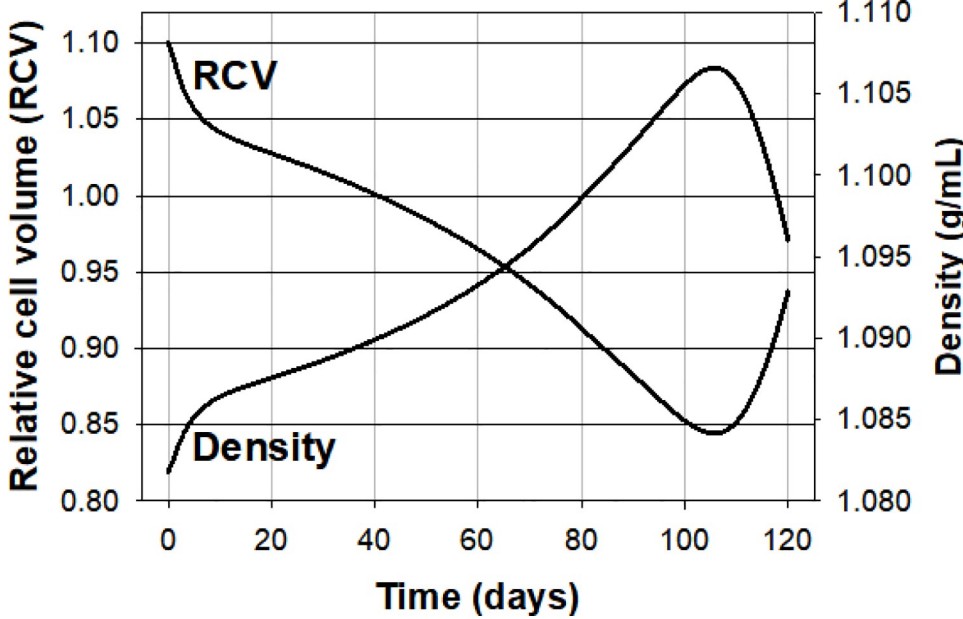

**Fig 3. Pattern of change in relative cell volume and density over a standardized 120 day lifespan period that best represents available evidence.** The lines join data points collected at daily intervals. The Lifespan model was run with the cell volume fraction set at 0.9 during the brief capillary transit period. "Restore medium" was set to YES to prevent carryover changes in medium concentrations during inter-transit periods. **RCV:** Relative cell volume, a convention adopted in RBC homeostasis models to report RBC volumes relative to a standardized value of 1 L/Loc attributed to a RBC defined with 0.75 Lcw/Loc, 0.25 LHb/Loc and 340 gHb/Loc (User Guide, Appendix). **Density:** Density profile, in g/ml. The initial condition of the cell was defined with Vw of 0.85 Lcw/Loc, CNa of 5 mM, CK of 145 mM, and a Na/K pump-mediated Na$^+$ efflux rate of -3.2 mmol/Loch, an approximate representation of the condition of a recently transitioned cell from reticulocyte to mature RBC with its full complement of haemoglobin set at 340 gHb/Loc. The parameter values used were: OS, 0.4s; PzCa, 70/h; PzA, 50/h (no significant differences between 30/h and 50/h for PzA, the range observed under on-cell patch clamp [19]); kCaP, 8e-6/min; kNaP, 3e-5/min; TNaP, 115200min; PzNa and PzK were set to zero. The RCV curve is used as a standardized reference (Ref) for analysing the effects of parameter variations in Figs 4 and 5.

variations thus allowing a detailed analysis of the mechanisms involved (Figs 4 and 5). The reader has immediate access to this "Ref" pattern by running the Lifespan*.jar model programme with the set defaults for initial conditions and parameter values.

Fig 3 shows the predicted changes in relative RBC volume and density in simulations aimed at defining the minimal set of parameter values able to provide full compliance with the experimental constraints. These values are listed in the legend of Fig 3. As explained in detail for single transits [1], realistic representations required high values for the cell/medium volume ratio (equivalent to high haematocrits) during capillary transits. All the simulations reported here were set with an open state (OS) duration of 0.4s and with one minute duration for inter-transit periods. For an OS of 0.4s a nominal calcium permeability (PzCa) through PIEZO1 channels of 70h$^{-1}$ rendered the results shown in Fig 3 for our reference condition.

The density distribution of normal human RBCs follows a Gaussian pattern with means around 1.085–1.095 g/mL [6]. Two conditions ought to be fulfilled in the model to comply with such a pattern. The first is that for most of the time the cells spend in the circulation their density should vary around the measured population mean. This implies a dominant period of gradual densification of "middle age" cells (~ between days 20 and 100 of circulatory age in the model, Fig 3), with younger and older cells in lower and higher density states, respectively, contributing lesser proportions of aging time. The second condition is that for the Gaussian pattern not to be significantly altered at the high density end, the contribution of the oldest

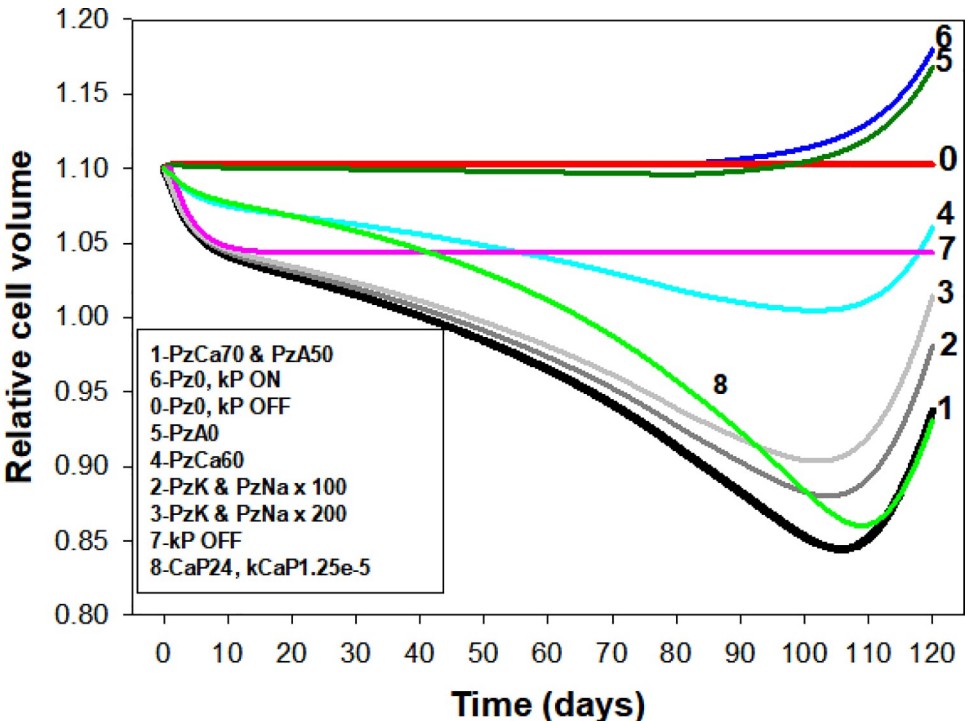

**Fig 4. Effect of changes in PIEZO1-mediated ionic permeabilities (Pz) and pump decay rates (kP) on the lifetime pattern of cell volume change. 1.** The **p**arameter values for Reference curve 1 (black) are as reported in the legend of Fig 3. **0:** With PzX = 0 and no pump decay the model computes a flat response over the full 120 days period demonstrating the robust stability of the Lifetime computations. **7:** With the PzX set as for the reference curve (curve 1) but with no pump decay (curve 7, kP OFF) there is no progressive dehydration-densification, only the early quantal dehydration reported in Fig 2. **6:** With PzX = 0 and pump decays set to ON (curve 6, Pz0, kP ON) there is no dehydration phase, only late hydration following delayed Na/K pump decay. **5:** Same as curve 1 but with PzA set to zero showing how extremely limiting the anion permeability can be to both initial and cumulative dehydration responses. **4:** Relatively minor reductions in PzCa from 70/h (Curve 1) to 60/h (curve 4) reduce initial and cumulative dehydration responses outside observed ranges. **2 & 3:** Large changes in PIEZO1-mediated $Na^+$ and $K^+$ permeabilities, curves 2 and 3, have relatively minor effects, mostly on the timing and magnitude of the late density reversal response, rendering the Lifespan model a poor predictor of their likely real values. **8:** Protocol identical to that of reference curve 1 but for a cell defined in the RS with a FCaPmax of 24 instead of 12 mmol/Loch; the increased pump strength reduced the extent of early dehydration and in order to approximate the reference dehydration pattern as shown it was necessary to increase kCaP from 8e-6 to 1.25e-5 $min^{-1}$.

cells undergoing density reversal must be limited both in their proportion and in the extent of density reversal, a powerful constraint on the model parameters controlling Na/K decline, onset time and rate of decline.

The volume curve (Fig 3) will be used as our reference for all further explorations of the parameter space (black curves in Figs 3–5, 7 and 8). The volume and density curves (Fig 3) are mirror images of each other as expected from the model-assumed approximation that the haemoglobin complement of each cell after maturation from the reticulocyte stage remains constant throughout its lifespan. Thus, changes in density reflect only variations in hydration state. The rate of volume change during the densification phase, between days 10 and 100 was about -0.2%/day (Fig 3).

The mechanism by which gradual decay in the activity of the calcium pump induces progressive dehydration operates by gradually weakening the calcium pump and so delaying the extrusion of the $Ca^{2+}$ gained over the brief PIEZO1 open state period during each capillary transit ([1], Fig 6C). The duration of elevated $[Ca^{2+}]_i$ states in successive capillary transits

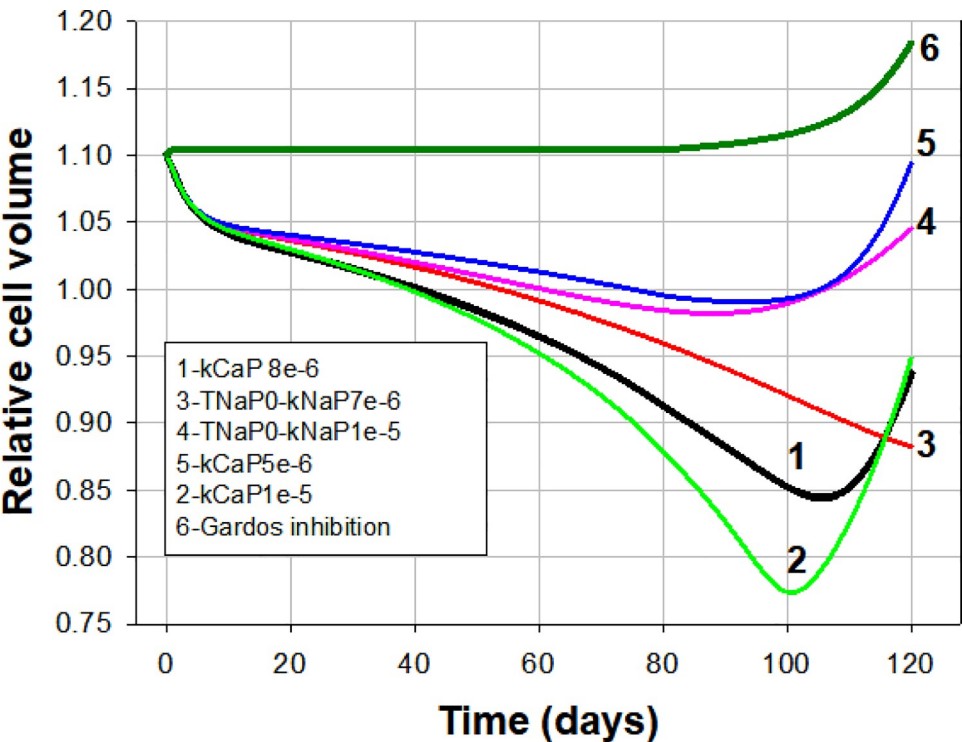

**Fig 5. Effects of pump decay rates and timings, and of Gardos channel inhibition on the predicted lifetime pattern of cell volume change.** Parameter changes are listed relative to reference curve 1. For the simulation shown in curve 6 the Gardos channel Fmax was changed from 30/h to zero in the Reference State. For curves 3 and 4 TNaP was set to zero and the decay rates to 7e-6/min and 1e-5, respectively. For curves 2 and 5 PMCA decay rates were set to 1e-5 and 5e-6, respectively.

increases progressively by the weakening PMCA thus increasing the extent of Gardos channel-mediated dehydration in successive transits ([1], Fig 6C–6E). This breaks the balance which halted dehydration progress during consecutive quantal transits under the assumption of constant pump strength in the quantal hypothesis tests (Fig 2). PMCA decay changes the quantal hypothesis stalemate from |W1| = |W2+W3+W4| to an increasing imbalance where |W2+W3+W4| > |W1| (Fig 5 in [1]).

The exponential decay rate of the PMCA found to comply best with observed densification patterns (Fig 3) predicted an age-determined distribution of PMCA Fmax activities for circulating RBCs in remarkable agreement with experimental results [12]. The best fit (Fig 3) followed the equation $y = y^{o*}\exp(-(8^*10^{-6})t)$. The y-values computed from this equation for the 120 days of our standard lifespan period, report the age–dependent Fmax decline of the model PMCA with a distribution whose statistical parameters can be compared to those of measured ones. Using the default PMCA Fmax value of 12 mmol/Loch for $y^o$ the resulting statistical parameters for the Ref PMCA decline function were: median, 6.12; mean, 6.64; SD, 2.69; CV 46% and skew of 1.2 which compares with measured skews of between 1.4 and 1.7 in RBCs from six different donors. Similar CV and skew values are obtained with different $y^o$ and kCaP pairs constrained to approximate the reference densification pattern, as for curve 8 in Fig 4, for instance. The similarity between measured and predicted CVs and skews suggests that the variation in PMCA activity in RBC populations from healthy adult donors is almost entirely age-related, and that its decay pattern was harnessed by evolution to extend the circulatory lifespan of RBCs.

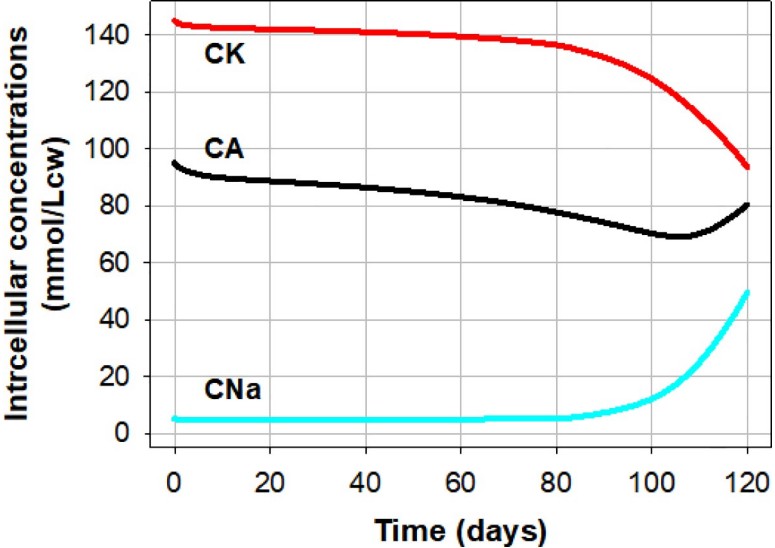

**Fig 6. Predicted lifespan changes in the intracellular concentrations of Na (CNa), of K (CK) and of diffusible anions A (CA) for the conditions of the Reference pattern (Fig 3).** Note that because PzNa and PzK were set at zero in the minimalistic representation of the reference pattern, the predicted changes shown in this figure result solely from the cumulative effects of periodic Gardos channel activation and Na/K pump decay.

As analysed in detail below, a delayed onset of an exponential decay in sodium pump activity was required for early permissive densification and late density reversal.

## Analysis of the effects of PIEZO1-mediated permeabilities and pump decay

When all PIEZO1-mediated permeabilities and pump decay rates are set to zero (Fig 4, curve 0) the model computes a flat response over the full 120 day period and $10^9$ model iterations demonstrating the noise-free robust stability of the Lifetime computations. Restoring pump-decay rates for both PMCA and Na/K pump, with PIEZO1 blocked (Fig 4, curve 6) the response remains flat until a late hydration stage resulting from delayed Na/K pump decay. This shows that PMCA decay on its own, without PIEZO1 activity, cannot generate gradual densification. Moreover, it never decays sufficiently to fail balancing $Ca^{2+}$ influx through the intrinsic $Ca^{2+}$ permeability of the RBC membrane. These results stress the absolute need to involve PIEZO1 mediation in the generation of the reference pattern. With pump decay rates set to zero (Fig 4, curve 7), the model predicts a minor dehydration over the first few days, followed by a flat volume response, the full extent of dehydration expected from the quantal hypothesis (Fig 2). Gradual decays in the activities of the calcium and Na/K pumps are therefore a necessary condition for generating the pattern of progressive changes in RBC volume during circulatory senescence (Fig 3). Taken together (Figs 2–4), these results show that both processes, quantal PIEZO1 activations and pump decays are necessary for shaping the observed pattern of lifespan RBC volume changes, neither sufficient on its own.

We investigate next the role of the anion permeability in the generation of the lifetime dehydration-densification pattern. A substantial increase in anion permeability had been detected associated to the deformation-induced increased $Ca^{2+}$ permeability in on cell patch clamp experiments [19,20], but it was not clear whether this was an incidental association of relevance to lifespan dehydration. With PzA set to zero in the lifespan model, dehydration was severely reduced (Fig 4, curve 5) showing an extremely limiting effect of the anion permeability, much more powerful than the one predicted for single capillary transits [1]. These results

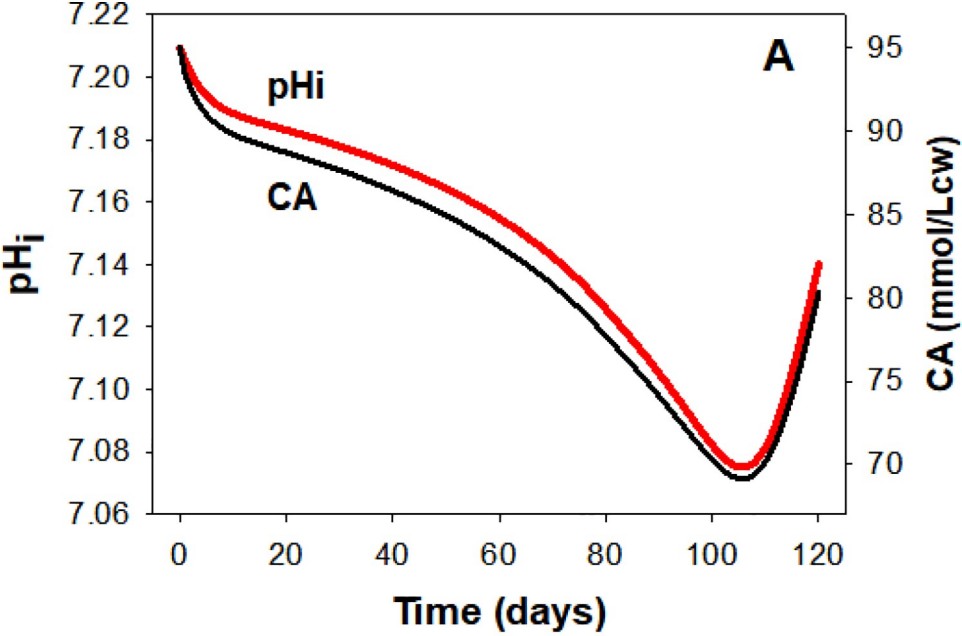

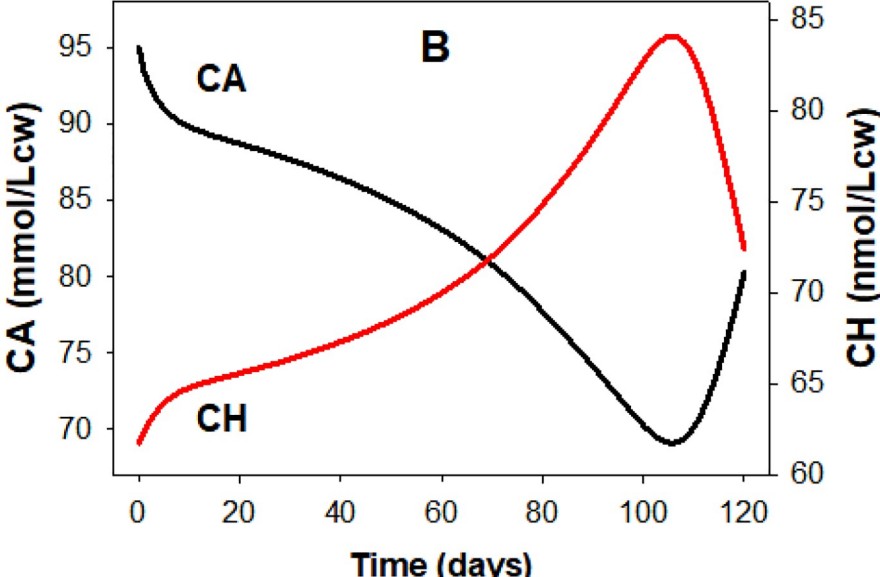

**Fig 7. Predicted lifespan changes in cell pH (pHi) and in the intracellular concentrations of permeant anions A⁻ (CA) and of H⁺ (CH) for the conditions of the Reference pattern (Fig 3).** Besides minor contributions from variations in haemoglobin buffering, most changes in CH are driven primarily by changes in the anion concentration gradient (rA) acting through the Jacob-Stewart mechanism causing rH to approach rA during periods between capillary transits (User Guide). At constant MA all anion gradient changes apply to CA, thus generating similar time-courses for CA and pHi (**Panel A**), or mirror image changes for CA and CH (**Panel B**).

show that increased anion permeability is a necessary companion to PzCa in the generation of the lifespan pattern (Fig 4, curve 1). The attribution of the increased PzA to PIEZO1 in the simulations is purely operational; its actual mediation remains an open question [20]. The model simulations revealed an additional unexpected role of the increased anion conductance,

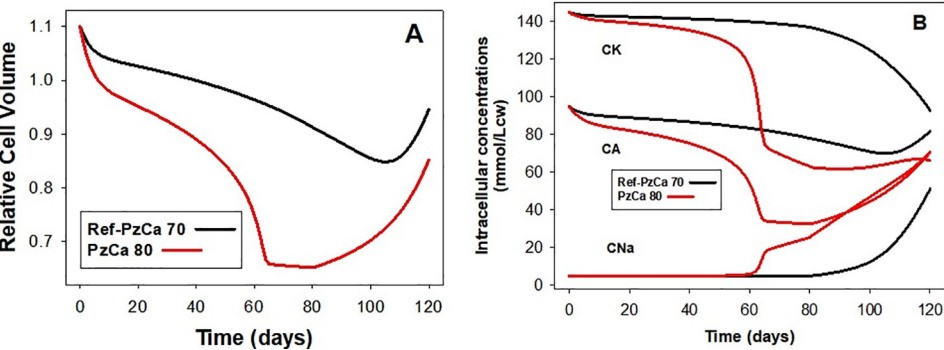

**Fig 8. Predicted changes in RBC volume (A), and in cell concentrations (B) of K(CK), of Na(CNa), and of permeant anions (CA) during hyperdense collapse induced by elevated PzCa.** The changes induced by setting PzCa = 80/h (red curves) are compared with those of the reference pattern with PzCa = 70/h (black curves).

as shown by comparing curve 7 with curves 5 and 6 in Fig 4. If PzA was not increased, whether or not in association with increased PzCa, the relatively rapid initial dehydration phase was absent. This result and additional tests showed that the relatively rapid early dehydration phase is largely PzA-dependent.

## Effects of PIEZO1-mediated sodium and potassium permeabilities

In sharp contrast to the powerful effects of PzCa and PzA on the kinetic pattern of the dehydration response, increasing PzNa and PzK from zero to over two orders of magnitude above ground sodium and potassium permeabilities had minor effects (Fig 4, curves 2 and 3), mostly on the timing and magnitude of the late density reversal response. The minor effects of large increases in PzNa and PzK render the Lifespan model a poor guide on the likely values of real PIEZO1-mediated Na$^+$ and K$^+$ permeabilities during capillary transits, the reason for setting their default values to zero in our minimalistic approach to the attribution of default values for the reference curve.

## Analysis of the effects of pump decay rates and Gardos channel inhibition

Blocking the Gardos channel suppresses fully the dehydration-densification component of the response (Fig 5, curve 6) stressing the critical role of this channel in the PIEZO1-mediated lifespan densification process. Gardos channel activity has been shown to marginally decline with RBC age [13]. Model tests incorporating declining Gardos Fmax within observed boundaries had no detectable effects suggesting that decay in Gardos channel activity has no functional role in RBC senescence.

By setting the Na/K pump decay to start from the beginning (TNaP = 0) the dehydration rate becomes substantially reduced whether late reversal is curtailed (Fig 5, curve 3) or preserved (Fig 5, curve 4). Multiple additional simulations testing different pump-decay rates showed that early Na/K pump inhibition, by allowing progressive net NaCl gains over KCl losses partially offsets the dehydrating effects of Gardos channel mediated KCl losses, keeping the dehydration profile below observed boundaries. This explains the requirement of a substantial delay in the onset of Na/K pump decay for the reference patterns to emerge (Fig 5, curve 1), and suggests a double evolutionary input in the pattern of age-related changes in Na/K pump activity: prevention of early decay and late onset of decay. These simulations, of course, do not rule out some minor levels of early decay; they only stress the absolute requirement of late decay for density reversal.

The rate of decline in Na/K pump activity found to approximate the observed extents of density reversal predict a decline in pump-mediated sodium efflux from about -3.2 mmol/Loch on day 80 to -2.5 mmol/Loch on day 120, a 22% flux decline (data from reference csv file). The real decline in pump Fmax, on the other hand, set at $e^{-kt}$, with k = $3 \cdot 10^{-5}$ (min$^{-1}$), is 82% within this 40 day period. The reason for the discrepancy is that the flux was being strongly stimulated by the increasing intracellular sodium concentration (Fig 6), in turn the result of sharp progressive Fmax pump decline. Therefore, flux comparisons performed without correcting for differences in intracellular sodium concentrations between RBCs of different ages would tend to underestimate the true magnitude of the Na pump decay, as noted by Cohen et al., [15].

Curves 2 and 5 in Fig 5 show the volume effects of a faster (Fig 5, curve 2) or slower (Fig 5, curve 5) PMCA decay rate relative to the reference curve 1. It can be seen that minor changes in PMCA decay rates can cause substantial displacements and inflexion variations relative to the profile of the reference curve, but the overall lifespan pattern is retained. These results stress once more how strongly the observed lifespan pattern constrains the values of pump-decay rates and the range of onset times for the Na/K pump decay (TNaP). The conditions represented by curves 2, 3 and 8 in Fig 4 and by curve 2 in Fig 5 compare to curve 1 in their entitlements to represent densification patterns compliant with reality.

The interesting insight arising from this analysis is that declines in pump activity, generally assumed to be unavoidable side effects of cumulative protein damage in the absence of biosynthetic renewal, suddenly acquire a critical role in extending the circulatory lifespan of RBCs by keeping them within the reduced volume range for optimal rheology, a role harnessed by evolution through a tight control of their decay patterns.

## Predicted lifespan changes in intracellular Na$^+$, K$^+$ and A$^-$ concentrations

Over the first 70–80 days, the model predicts a slow quasi linear fall in CK in excess of the much slower CNa rise, with the CA decline following the trend set by the sum of the net cation concentration changes (Fig 6). After about day 80 the trend rapidly reverses causing a delayed CA reversal as soon as sodium gains exceed potassium loses. These model-predicted trends are in full agreement with all previous measurements of age-attributed sodium and potassium content changes in RBCs.

## Predicted changes in intracellular pH (pHi) and proton ion concentration (CH) and their relation to the parallel changes in intracellular concentrations of diffusible anions (CA)

The kinetics of the pHi changes follows closely that of the change in intracellular anion concentration (Fig 7A) as expected from an anion ratio driven process. As explained before [11,21–23] and in extra detail in the User Guide, the Jacob-Stewart mechanism (JS) drives the rH ratio ($[H^+]_i/[H^+]_o$) to match the rA ratio ($[A^-]_o/[A^-]_i$), a match reflected in the mirror images of the CH and CA curves of Fig 7B. The large capacity of the JS mechanism ensures that the infinitesimal changes in anion concentration ratio generated during each transit are rapidly approximated by the proton concentration ratio during inter-transit periods. Here again, isolated measurements of pHi in density-segregated RBCs fully support gradual cell acidification as predicted by the model (Fig 7A).

## Statistical compliance of predicted volume changes with cell age

The default settings of the csv-file outputs in the lifespan model were designed to report changes in variable values at daily intervals over a standard 120 day period. Each row in the csv

file predicts the values of all system variables for a particular day. The condition of the RBC at time = 0 is defined by the entries in the Reference State that define the initial condition of the modelled system. Each column reports how each variable changes daily throughout the 120 days lifespan offering a distribution of values open to statistical comparisons with measurements. Measured variations in RBC properties combine production line diversity, as RBCs emerge from the bone marrow, with changes caused by circulatory aging whereas model-predicted variations report only changes caused by circulatory aging. It follows that the predicted coefficient of variation of the columns representing day cohorts has to be smaller than that of the corresponding measured variable in real samples. The haematological variables for which we have extensive and reliable measurements are RBC volume, haemoglobin contents and haemoglobin concentration [5]. Of these, the only one allowing comparison between predicted and measured values is RBC volume, because the haemoglobin content of the model RBC, fixed "at birth" in the Reference State, is assumed to remain constant throughout the RBC lifespan. The mean and standard deviation values of the relative cell volume distribution for the cell represented by the reference pattern (Fig 3) were 0.956 and 0.069, respectively, rendering a coefficient of variation of 7.2%, well below measured values around 12–13% [5]. To the extent these predicted values approximate reality they suggest that about half the volume variations in measured RBC samples are the result of aging-induced changes in RBC hydration state.

## Hyperdense collapse and PIEZO1-mediated calcium permeability

A relatively minor reduction in PzCa, from 70/h to 60/h (Fig 4, curve 4) proved sufficient to prevent dehydration reaching the observed range. On the other hand, increasing PzCa above 70/h eventually overwhelmed the declining $Ca^{2+}$ extrusion capacity of the PMCA and triggered a sharp hyper-dense collapse, an extreme dehydration response (Fig 8A). The model predicts partial volume and density recovery post-collapse (Fig 8A) resulting from net NaCl and fluid gains following progressive Na/K pump decay, suggesting a potential for full lifespan survival of the cells (Fig 8A and 8B). It remains an intriguing open question whether such a sequence may, very occasionally, occur in vivo. Volume collapse takes a few days to evolve and much longer to recover so that the probability that RBCs could bypass spleen sinusoids during the hyperdense stages seems remote.

The results in Figs 4 and 8 show that the product OSxPzCa navigates a narrow range of variation between failing to densify sufficiently and hyperdense collapse. The $70h^{-1}$ PzCa value at an OS of 0.4s used for the Ref simulations (Fig 3) represents an increase of around three orders of magnitude over the basal calcium permeability of the RBC membrane, of $0.05h^{-1}$, the value found to fit the observed physiological rates of PMCA pump-leak turnover [24]. In more comparable permeability units this represents an effective increase in RBC membrane calcium permeability from $\sim 10^{-9}$cm/s, one of the lowest in nature, to $10^{-6}$cm/s. The OSxPzCa product represents a complex mix combining OS duration, number of PIEZO1 channels per cell and the intrinsic $Ca^{2+}$ permeability of each channel as influenced by cell and medium sodium and potassium concentrations [25]. In vivo, some of these components will vary from transit to transit, OS with speed of flow, and number of PIEZO1 channels activated by extent of deformation. Ultimately, according to the model, it is the aggregate of $Ca^{2+}$ loads that PIEZO1 activation induces during each capillary transit (Fig 5 in [1]), averaged over many transits, what counts for shaping the Lifespan pattern.

## Effects of RBC volume fraction during capillary transits on Lifespan patterns

The biphasic volume change patterns of RBCs during single capillary transits proved to be almost identical when simulated with cell volume fractions (CVFs) of 0.9 during their brief

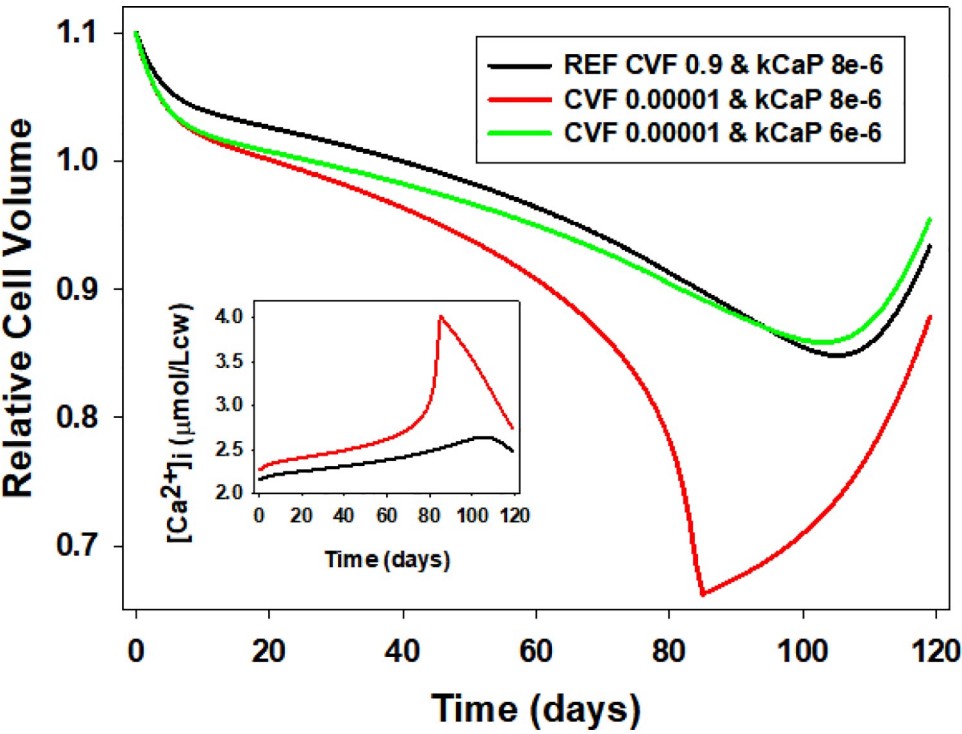

**Fig 9. Effects of high and low cell volume fractions (CVF) during capillary transits on the lifespan patterns of changes in cell volume and [Ca²⁺]ᵢ.** Reference pattern (black) modelled with CVF of 0.9 during capillary transits. Test curves (red and green) modelled with CVF of 0.00001 throughout transit-intertransit periods. PMCA decay rates (kCaP), as indicated in the figure (in min⁻¹). Note how the reduction in kCaP from 8e-6 (red) to 6e-6 (green) prevents hyperdense collapse at CVF 0.00001 and restores pattern towards Reference curve (black). **Inset:** [Ca²⁺]ᵢ values (CCa2 +) recorded at the end of PIEZO1 open states on the last model iteration of each day (col 19 of the transit csv files). Note the increasing divergence between REF (black) and low-CVF (red) curves leading to hyperdense collapse. Late reversal of [Ca²⁺]ᵢ results from terminal rehydration of the cells.

capillary transits, or of 0.00001 throughout both transit and inter-transit periods [1]. However, when the effects of the two alternative transit CVF values were compared on the lifespan model, under identical conditions and parameter values, the red cell volume of the low-CVF cell slowly drifted towards a late hyperdense collapse (Fig 9, red). At high CVFs the transfer of calcium from medium to cells was bound to decrease the medium calcium concentration significantly more than at low CVFs. Cell calcium gains during capillary transits at low CVFs were therefore expected to be slightly larger than those at high CVFs under comparable condition. These differences were too insignificant to cause noticeable volume changes on single transits [1], but had the potential to accumulate and overwhelm the Ca²⁺ extrusion capacity of a PMCA decaying at the REF-set rate when balancing the calcium gains of a low-CVF cell, as proved to be the case (Fig 9). The inset of Fig 9 shows the expected trend of change in [Ca²⁺]ᵢ as recorded at the end of the last capillary transit each day for low-CVF conditions (red) and for REF conditions (black). The restoration of comparability between high- and low-CVF conditions when the rate of PMCA decay at low-CVF is reduced (Fig 9, green) confirms that the discrepant volume response can be entirely attributed to differences in cumulative calcium gains. This outcome then justifies the choices made of modelling capillary transits and lifespans with the more realistic high-low CVF transitions between capillary and systemic circulation using the Restore Medium subroutine [1].

## Iirreversibly sickled cells (ISCs): fast track lifespan, hyperdense collapse, and terminal density reversal

The condition of the collapsed cells resembles that of the subpopulation of sickle cells known as irreversibly sickled cells (ISCs) [26–28]. These cells originate from a subpopulation of stress reticulocytes, have an extremely short, 4–7 day lifespan, dehydrate rapidly by a calcium-dependent fast-track mechanism on egress from the bone marrow [21,29], persist in the systemic circulation for a few days in a hyperdense condition, terminally gaining NaCl and rehydrating back to a low density state [6]. In their hyperdense state, ISCs are responsible for vaso-occlusion leading to multiple organ infarctions, pain crisis, and for most of the clinical symptoms of sickle cell disease (SCD) irrespective of their proportion in the circulation [30,31]. Early spleen infarction in infancy leads to functional asplenia [32] and prevents effective clearance of hyperdense ISCs in SCD patients.

The potential for hyperdense collapse exposed by the lifespan model prompted a search for conditions which could emulate the circulatory trajectory of ISCs. Fig 10 shows a predicted lifespan for a seven-day ISC. The open state attributed to PIEZO1 as Psickle [33], the permeability pathway generated by the interaction of deoxy-haemoglobin S polymers with the inner membrane surface, was set at 30 seconds emulating the observed extended periods of elevated calcium permeability in deoxygenated conditions that might be expected during inter-transit periods in the venous systemic circulation [34]. Sickling follows a ~ 40th power dependence on the concentration of deoxy-haemoglobin S (deoxy-HbS) [35,36]. Yet sickle reticulocytes, still far short of the full HbS complement of the mature red cell, and with a far lower haemoglobin concentration, showed a sharp and reversible permeabilization response to $K^+$, $Na^+$ and $Ca^{2+}$ on deoxygenation [29], a critical finding yet to be explained.

It was shown before that fast track dehydration of ISCs could be induced by a sequential, self-reinforcement mechanism involving Psickle/PIEZO1 [33], Gardos channels and the K:Cl cotransporter KCC3 ([21,29,37,38]. Sickling-induced $Ca^{2+}$ influx activates Gardos channels during deoxy-states leading to RBC dehydration and acidification. Acidification, in turn, stimulates further KCl loss, dehydration and acidification via KCC3 [38,39], a transporter highly expressed in Hb S stress reticulocytes, speeding dehydration by remaining active during oxy-states in the circulation. The current results (Fig 10) show that in RBCs with an enhanced PIEZO1/Psickle calcium permeabilization response, as observed with deoxy-Hb S reticulocytes [29], the PIEZO1-Gardos channel partnership alone can emulate the typical lifetime timecourse of ISCs [26]. The relative roles of PIEZO1 and KCC3 in ISC fast track dehydration remain to be elucidated, but if PIEZO1 turns out to be such a potent component of the fast-track response as the current results suggest (Fig 10), its erythroid cell expression becomes a prime therapeutic target in sickle cell disease.

## Spontaneous inactivation of PIEZO1 and of Psickle channels

The 30s open state attributed to Psickle in the simulation of Fig 10 was meant to represent the active condition of this channel during inter-transits in the venous systemic circulation, implying a condition of suspended or delayed inactivation supported by a variety of experimental observations. RBCs from heterozygote HbSA subjects with largely normal membrane properties, when exposed to deoxy-sickling pulses lasting hours remained in a high $Ca^{2+}$ permeability state for the full duration of the pulses, recovering fully on reoxygenation [40]. This was also shown to be the response of HbS RBCs during extended sickling periods in vitro [33,34]. Thus, inactivation appears to be reversibly suspended for the duration of the sickled condition. When PIEZO1 channel activation is elicited in HbA RBCs by contact with electrodes, inactivation also appears disrupted or profoundly delayed [19,41]. On the other hand, calcium signals

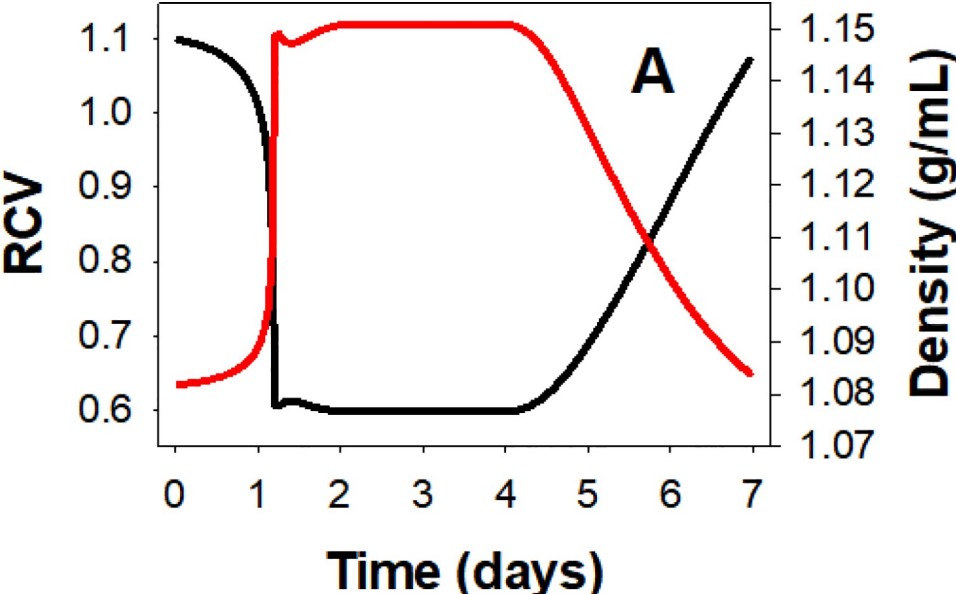

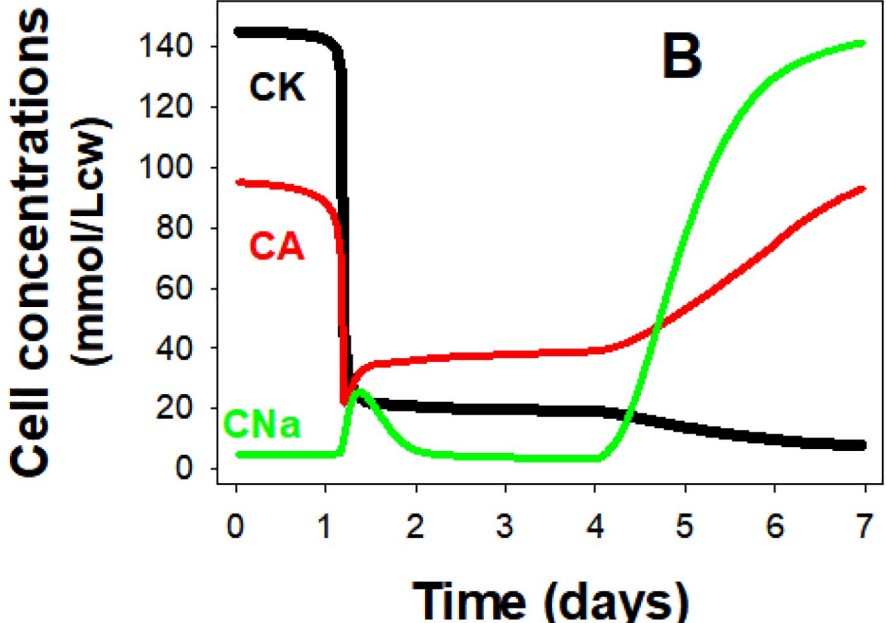

**Fig 10. Simulating the lifespan of a seven day irreversible sickle cell (ISC), showing the predicted changes in RCV (black) and density (red), (panel A), and in the cell concentrations (panel B) of K(CK), of Na(CNa), and of permeant anion (CA).** The parameters used were: Lifespan duration, 168h (7 days); Data output periodicity: 60m; kNaP, 2e-3/m; kCaP, 5e-5/m; TNaP, 5760m (4 days); PIEZO1 open state, 30s; PzCa, 5.7/h; PzA, 50/h; PzNa, 0; PzK, 0.

recorded from Fluo-4-loaded RBCs traversing microfluidic constrictions under pressure regimes that ensure transit times approximating those in vivo, conditions assumed to preserve normal membrane-cytoskeletal connectivity, showed reversible fluo-4 signals reporting elevated $[Ca^{2+}]_i$ only during transits, the expected response of PIEZO1 channels with preserved spontaneous inactivation ([41] their Fig 3).

Taken together, these disparate observations suggest that local disruption of the normal structural connections between the plasma membrane and cytoskeleton, either by HbS polymers protruding through the cytoskeletal mesh from the inside of the membrane or by electrode contact or suction, affect subpopulations of Psickle/PIEZO1 channels [42] somehow disturbing configurations required for spontaneous inactivation functionality, a hypothesis originally advanced by Gottlieb et al., [43] from whole cell and patch recording studies.

Interactions between the molecular inactivation machinery of PIEZO1 channels and the changing cortical cytoskeleton of cells during the reticulocyte-mature-RBC transition may also hold the answer to the unexplained nature of the high cation leak pathways of reticulocytes. The mean $Ca^{2+}$ and Na/K pump-leak turnover of reticulocytes is about ten-fold higher than that of mature RBCs [44,45]. The nature of the $Ca^{2+}$, $K^+$ and $Na^+$ leak pathways remains a mystery [21,29,38]. An intriguing possibility worth future investigation is that amidst the profound cytoskeletal-membrane remodelling processes taking place after erythroid cell enucleation the high cation leaks are mediated by open or partially open PIEZO1 channels, roughly estimated in the hundreds per cell at the reticulocyte stage [42]. As spontaneous inactivation functionality becomes established with progress towards the membrane-cytoskeletal configuration of mature RBCs so does the declining magnitude of the leak fluxes.

These considerations highlight a set of open questions arising from the current lifespan investigation implicating PIEZO1 inactivation as the target of interest in pathologies affecting RBC hydration and suggest altered membrane-cytoskeletal interactions as the focus of attention for explaining the molecular mechanism of disrupted PIEZO1 inactivation [46].

## Discussion

We approached the investigation of the mechanisms shaping the lifespan changes of human RBCs, a subject inaccessible to direct experimentation, by applying a dedicated extension of the core red cell model RCM [1], the Lifespan model (Fig 1). We started by questioning the nature and range of RBC responses to be expected from deformation-induced PIEZO1 activation during single capillary transits [1], and followed this up in this paper with a systematic exploration of the dynamic combinations of homeostatic processes that could deliver the documented patterns of change throughout the $\sim 2 \bullet 10^5$ transits RBCs experience during their long circulatory lifespan.

A first result was the demonstration of the inadequacy of repeated PIEZO1 channel activations during capillary transits to generate long-term progressive RBC densification on their own (Fig 2), their cumulative power fading rapidly within days to minimal densification levels, thus ruling out the quantal hypothesis [10,47]. The mechanism that finally emerged involved a complex interplay among a quartet of membrane transporters (PIEZO1, Gardos channels, PMCA and Na/K pumps) involving a tightly defined decay pattern for the pumps (Figs 3–5) and additional modulating influences by all other membrane and homeostatic components of the RBC (Figs 6–8), reported and analysed in detail in Results and Analysis.

Looking back at the variety of conditions which failed on compliance with the established densification-late-reversal pattern (Figs 4 and 5), there are clear indications that RBC volume stability and hence adequate rheological performance throughout extended lifespans could also be attained by other, apparently simpler alternatives. Playing with the model unconstrained by facts, one alternative emerged which, surprisingly, was also tightly constrained, that of a RBC without PIEZO1 channels (PzX = 0) and without Gardos channels (PKGardos-Max = 0 in RS), but with a well balanced pattern of pump decays (in $min^{-1}$, kCaP = $8^*10^{-5}$; kNaP = $1^*10^{-6)}$), a simple pump-controlled duet mechanism in which the opposite swelling-shrinking effects elicited by decaying Na/K and PMCA pumps, respectively, remain well

balanced throughout. A RBC like this, free from sudden permeability changes during capillary transits by the absence of PIEZO1 channels, and exempt from hyperdense collapse threats (Fig 8) by the absence of Gardos channels could sustain excellent volume stability and optimal rheology throughout extended lifespans at slightly lower metabolic cost that a quartet cell exposed to periodic PMCA stimulation. This prompted the question of what favoured or determined the evolution of the quartet mechanism in human RBCs.

There is no strong argument for a selective preference of quartet over duet mechanisms or other alternatives. In different species the universals of optimal economy and rheology providing extended lifespans are realized with very different strategies, typical of adaptive solutions on the go operating on pre-existing conditions [48]. There are well documented instances of species whose RBCs lack Gardos channels [49], have kinetically diverse and even absent Na/K pumps [50,51] and varied Na/K concentration ratios [52], have different constellations of membrane transporters controlling RBC volume and homeostasis [53], differ substantially between foetal and mature RBCs [54], have completely different cytoskeletal structures, with and without vestigial organelle retention [55,56]. There is no information available yet on how widespread the presence of PIEZO1 channels is in RBCs from different species, and therefore on how central its role may be in the dynamics of capillary circulation. So far, the only documented constant in all species appears to be the calcium pump around which all the different lifespan strategies of RBCs evolved.

Mutant PIEZO1 channels in hereditary xerocytosis (HX) were found to exhibit a number of kinetic abnormalities the most prominent of which was a marginally reduced inactivation kinetics following brief stretch-activation pulses [57]. Simulations with the Lifespan model show how relatively small inactivation delays can lead, after myriad capillary transits, to profound RBC dehydration approaching hyperdense collapse (Fig 8) with similarities to observed alterations in HX RBCs [58]. Protection against severe falciparum malaria, the reason for the persistence of many genetic mutations affecting RBC hydration in human populations, is contributed by two common conditions: anaemia and the presence of subpopulations of dense RBCs which falciparum merozoites fail to infect [59], thus preventing the build up of the high parasitaemias required to cause cerebral malaria, the main malaria killer [60,61].

The lifespan model opens the way for further in depth studies on the changes in RBC homeostasis during circulatory senescence. With growing information databases on the genetics and pathologies associated with the transport systems that control the lifespan of RBCs, model versions encoding known or hypothesized abnormalities of those transporters may become useful tools in furthering the understanding of the pathophysiology and clinical manifestations of the diseased conditions. Within the mathematical-computational framework of the red cell and lifespan models applied here for human RBCs, components can easily be modified and adapted to explore RBC homeostasis, circulatory dynamics and lifespan strategies across species paving the way for future studies on the comparative circulatory biology and pathology of RBCs, and on the effects of alternating oxy-deoxy capillary transits.

## Acknowledgments

The authors are grateful to Serge L. Y. Thomas and Daniel J. Lew for helpful comments and suggestions on the material contained in these papers and in the RCM User Guide.

## Author Contributions

**Conceptualization:** Simon Rogers.

**Formal analysis:** Virgilio L. Lew.

**Funding acquisition:** Virgilio L. Lew.

**Methodology:** Virgilio L. Lew.

**Software:** Simon Rogers.

**Visualization:** Simon Rogers.

**Writing – original draft:** Virgilio L. Lew.

**Writing – review & editing:** Simon Rogers, Virgilio L. Lew.

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
