## [Decision Letter · Decision Letter 0]

22 Sep 2020

Dear Dr Lew,

Thank you very much for submitting your manuscript "PIEZO1 and the mechanism of the long circulatory longevity of human red blood cells" for consideration at PLOS Computational Biology. As with all papers reviewed by the journal, your manuscript was reviewed by members of the editorial board and by several independent reviewers. In light of the reviews (below this email), we would like to invite the resubmission of a significantly-revised version that takes into account the reviewers' comments.

Unfortunately I am having no success in finding reviewers willing to review the companion paper to this one. Although I had hoped to find reviewers willing to review both paper, that may not be possible. I am send the "major revisions" decision on this paper to give you a jump start on addressing issues that may be relevant to both papers. And I am still working to find reviewers for the companion paper. Please bear with me.

We cannot make any decision about publication until we have seen the revised manuscript and your response to the reviewers' comments. Your revised manuscript is also likely to be sent to reviewers for further evaluation.

Sincerely,

Daniel A Beard

Deputy Editor

PLOS Computational Biology

Daniel Beard

Deputy Editor

PLOS Computational Biology

Reviewer's Responses to Questions

**Comments to the Authors:**

Reviewer #1: The manuscript proposed a very interesting question about how red cell volume changed in response to Piezo1 activation during capillary transit. In this case, computational approach was used to simulate red cell volume upon piezo1 activation and demonstrated that cell volume increased following by shrinkage during capillary transit and the magnitude of such volume change was small.

Comments

1) The kinetic of such process is not clear. For example, start from time 0 where red cells enter capillary, how long it takes to activate piezo1 channel and what is the timescale in other related signaling pathways that eventually lead to cell volume change (figure 4)? This kinetic analysis is critical to understand the question proposed by the authors.

2) It is not clear to me how the magnitude of volume change was calculated. Doe it based on the number of piezo1 channels on the cell membrane?

In terms of the second manuscript entitled “PIEZO1 and the mechanism of the long circulatory longevity of human red blood cells”, the description was comprehensive but it would be better if it could be more focused. For example, it would be interesting to focus on quantitatively how decay in different channel activities that regulated cell volume contribute to red cell longevity and what were the relative roles of each channel in this process.

Reviewer #2: I would like to thank the authors for taking time to develop a tool that is available for testing as an open-source software. I consider myself as a potential user of the software to try to design experiments of my own and try to predict their possible outcome. It is an interesting instrument that has to be tested. Experimental biophysics is in need of such modelling approaches if they prove valuable (it needs substantially more time to tell).

However, some statements, such as a definition of initial condition as “a collection of RBC clones in plasma-like medium” make researchers involved in actual experiments worry already now. What we see suggests that the cells are no “clones” at all, and are not equipped with the same number of channels and pumps. As a result, we are dealing with a collection of sub-populations of cells with different properties, that include, but are not reduced to cell age. These sub-populations vary in abundance and maximal level of activity PIEZO1 channels, responsiveness to mechanical and chemical stressors and with different activity/abundance of PMCAs. We regularly observe “non-responders to stretch, making 10-40% of all cells. PMCA decay is not a continuous process as the enzyme, when undergoing cleavage by calpain, changes its activity in a “quantum” way, going from uncontrolled up-regulation in Vmax to its complete inactivation or loss, or incorporation into the inside-out vesicles (in cells of patients with sickle cell disease this process is very pronounces, as the author of this paper has successfully demonstrated a while ago). How does the model address this process, would it change a lot in the outcome?

When looking at the data in Fig 4,5,9 the changes in ion content occur slowly over time. I would expect them to occur for different cells at different time points and indeed (as shown in the examples of 1 and 4 in Fig 4) depend a lot of Ca2+ permeability for different cells. We have seen that free Ca2+ levels are not increase in the densest RBC population, but are very high in some cells (presumably those undergoing volume reversal) in the light RBC fraction. This is in contradiction to the inset of Fig 9 and with the notion that terminal swelling will increase the membrane surface tension opening the PIEZO1. Can the model predict how many cells will be found at the terminal swelling stage in the whole population? Is there a “probability assessment” possible with it?

How is a membrane loss kinetics component addressed in the model?

How is the amount of bound water (mainly Hb-bound) that cannot participate in diffusion or volume regulation integrated into the model? This is a substantial fraction that changes with the changes in RBC volume.

**Have all data underlying the figures and results presented in the manuscript been provided?**

Reviewer #1: Yes

Reviewer #2: Yes

PLOS authors have the option to publish the peer review history of their article (what does this mean?). If published, this will include your full peer review and any attached files.

Reviewer #1: No

Reviewer #2: **Yes: **Anna Bogdanova
---

## [Decision Letter · Decision Letter 1]

6 Nov 2020

Dear Dr Lew,

We are pleased to inform you that your manuscript 'PIEZO1 and the mechanism of the long circulatory longevity of human red blood cells' has been provisionally accepted for publication in PLOS Computational Biology.

Congratulations on a tour-de-force and insightful modeling study. Please note that Reviewer 2 offers a final important point that you should consider addressing in your finalization of the paper.

Best regards,

Daniel A Beard

Deputy Editor

PLOS Computational Biology

Daniel Beard

Deputy Editor

PLOS Computational Biology

Reviewer's Responses to Questions

**Comments to the Authors:**

Reviewer #1: the manuscript is improved after revision. i have no further comments

Reviewer #2: Dear colleagues, thank you for addressing all the issues I was raising in my review. I only have one statement to make, and that is about the bound water. According to the recent work, 20% of water in RBCs are present as a part of Hb hydration shells, and, therefore, belong to the "bound water" pool to say nothig about ions and other molecules (see Latypova et al. J. Chem. Phys. 153, 045102 (2020); doi: 10.1063/5.0016437 153, 045102). With all my respect to Max Peruz, these are the experimental data we cannot ignore. So, I would like to ask how would these findings affect your model. It seems to me that the cytosol is more of a colloidal state than of a liquid. You will immediately notice it when trying to dissolve 35g of Hb in PBS... it is a gel.

**Have all data underlying the figures and results presented in the manuscript been provided?**

Reviewer #1: None

Reviewer #2: Yes

PLOS authors have the option to publish the peer review history of their article (what does this mean?). If published, this will include your full peer review and any attached files.

Reviewer #1: No

Reviewer #2: **Yes: **Anna Bogdanova

---

## [Editor Report · Acceptance letter]

13 Feb 2021

PCOMPBIOL-D-20-01636R1 

PIEZO1 and the mechanism of the long circulatory longevity of human red blood cells

Dear Dr Rogers,

I am pleased to inform you that your manuscript has been formally accepted for publication in PLOS Computational Biology. Your manuscript is now with our production department and you will be notified of the publication date in due course.

With kind regards,

Alice Ellingham
